# Contextual Position Encoding:
## *Learning to Count What's Important*

## Abstract

The attention mechanism is a critical component of Large Language Models (LLMs) that allows tokens in a sequence to interact with each other, but is *order-invariant*. Incorporating position encoding (PE) makes it possible to address by position, such as attending to the $i$-th token. However, current PE methods use token counts to derive position, and thus cannot generalize to higher levels of abstraction, such as attending to the $i$-th sentence. In this paper, we propose a new position encoding method, *Contextual Position Encoding* (CoPE), that allows positions to be conditioned on context by incrementing position only on certain tokens determined by the model. This allows more general position addressing such as attending to the $i$-th particular word, noun, or sentence. We show that CoPE can solve the selective copy, counting and Flip-Flop tasks where popular position embeddings fail, and improves perplexity on language modeling and coding tasks[1].

## 1 Introduction

Many common data sources such as text, audio, code, and timelines of events are ordered sequences. When processing such sequences, the ordering information is clearly critical. In the case of text, position information is vital not only for decoding meaning between words, but is necessary at every scale, such as the sentence and paragraph level. The Transformer architecture, which is the main backbone of current Large Language Models (LLMs), relies on the attention mechanism (Bahdanau et al., 2014) that inherently lacks ordering information and treats sequences as sets. Thus, it is necessary to have an additional mechanism for encoding position information. Position encoding (PE) (Collobert and Weston, 2008; Sukhbaatar et al., 2015) achieves this by assigning an embedding vector to each position and adding that to the corresponding token representations. Position itself can be measured in two ways: absolute PE that counts tokens from the start of a sequence, and relative PE that counts backward starting at the current token. PE methods have become an integral part of LLMs with several proposed variations of these basic themes (Dufter et al., 2022).

One common feature of existing PE methods is the use of tokens as the unit of measurement. However, a token is a variable unit that can be a whole word, or part of it, or even a character depending on the tokenization method. For Byte-Pair Encoding (BPE) tokenization (Sennrich et al., 2016), a word can be 1 or many tokens depending on the word itself. This position variance increases for more abstract elements like a sentence, which can have from ten to hundreds of tokens. Therefore token position is not suited for general position addressing such as finding the $i$-th word or sentence.

In order to tie position measurement to more semantically meaningful units such as words, or sentences, one needs to take context into account. However, this is impossible with current PE methods as position addressing is computed independently of the context, and then later merged with context addressing. We argue that this separation of the position and context addressing is the core problem, and instead we propose a new PE method that integrates context and position addressing together. In particular, we are interested in position encoding that is context dependent, so it can represent various levels of position abstraction at the same time, from token positions to sentence positions. This way, it is possible for example to use token positions to attend to the previous few tokens, while using sentence positions to attend to previous sentences for better understanding of the current sentence. We call our method *Contextual Position Encoding* (CoPE).

---

[1]The code is available at https://anonymous.link.com

Figure 1: **Contextual Position Encoding (CoPE).** Standard position encoding methods such as Relative PE are based on token positions. In contrast, CoPE computes gate values conditioned on the context first, then uses that to assign positions to tokens using a cumulative sum. This allows positions to be contextualized, and represent the count of different units like words, verbs or sentences. CoPE operates on each attention head and so can attend to different position types on each. In this example, attending to the last sentence using Relative PE is challenging, and the best it can do is a decaying attention ("recency bias"). CoPE can count the sentence endings and simply attend to position "0".

CoPE first determines which tokens to count using their context vectors. Specifically, given the current token as a query vector, we compute a gate value for each previous token using their key vectors. Then we aggregate those gate values to determine the relative position of each token with respect to the current token, as shown in Fig. 1. Unlike token positions, this contextual position can take fractional values, thus cannot have an assigned embedding. Instead, we interpolate embeddings that are assigned to integer values to compute position embeddings. Like the other PE methods, these position embeddings are then added to the key vectors, so a query vector can use them in the attention operation. Since contextual position can vary from query-to-query and layer-to-layer, the model can simultaneously measure distances in multiple units.

We first apply CoPE to several toy tasks: symbolic counting, selective copying and the Flip-Flop task, where it outperforms token-based PE methods, especially in the case of out-of-domain generalization. To test real-world applicability, we use a language modeling task on Wikipedia text where we show CoPE also leads to better performance. The same performance gain is also observed when trained on code. Further, we perform pre-training of 1.4B models from scratch and find that CoPE significantly improves perplexity, and leads to better accuracy on word counting task.

## 2 BACKGROUND ON POSITION ENCODING

The core of the attention mechanism is a softmax operation over tokens in a sequence (Bahdanau et al., 2014). Let $\{x_1, \ldots, x_T\}$ be a sequence of input tokens, and $\{\mathbf{h}_1, \ldots, \mathbf{h}_T\}$ be their hidden representations. The query $\mathbf{q}_i$, key $\mathbf{k}_i$ and value $\mathbf{v}_i$ vectors are built through linear transformations of $\mathbf{h}_i$. The attention outputs $\mathbf{o}_i$ for every $i$-th token are

$$\mathbf{o}_i = \sum_j a_{ij} \mathbf{v}_j \quad \text{where} \quad a_{ij} = \text{Softmax}(\mathbf{q}_i^\top \mathbf{k}_j).$$

This attention operation is invariant to position information $j$, so it becomes necessary to have an additional position encoding (PE) mechanism (Sukhbaatar et al., 2015). PE methods can be categorized into two main groups: absolute and relative. The absolute PE simply adds a vector representing an absolute position $j$ to the hidden states, usually after token embedding: $\mathbf{h}_j \leftarrow \mathbf{h}_j + P(j)$. Here $P(i)$ can be implemented by an embedding layer that assigns a unique learnable vector $\mathbf{e}[i]$ to each position value $i$. Alternatively, $P(i)$ can be a fixed mapping that uses sinusoidal functions with different frequencies (Vaswani et al., 2017).

Relative PE (Shaw et al., 2018) depends on the token position $j$ that is being attended to, in addition to the current token $i$. Therefore, it has to be implemented within the attention layer

$$a_{ij} = \text{Softmax}(\mathbf{q}_i^\top (\mathbf{k}_j + P(i - j))).$$

Here we added it to only the key vectors, but there are other variations. Again, $P$ can be an embedding layer so we have a learnable vector for each position:

$$a_{ij} = \text{Softmax}(\mathbf{q}_i^\top (\mathbf{k}_j + \mathbf{e}[i - j])). \tag{1}$$

Fixed functions can also be used, such as in RoPE (Su et al., 2024). Now, we can view the $\mathbf{q}_i^\top \mathbf{k}_j$ term as context-addressing because it depends on what the $x_j$ token actually is, and view $\mathbf{q}_i^\top \mathbf{e}[i - j]$ as position-addressing since it solely depends on position information of $x_j$. Although many different position encoding methods have been proposed (see Dufter et al. (2022) for a survey), with most focusing on improving efficiency, they are all based on token positions.

## 3 MOTIVATION FOR CONTEXTUAL POSITION ENCODING

### 3.1 STANDARD POSITION ENCODING FAILS ON SIMPLE TOY TASKS

Here we analyze a simplified attention mechanism and a toy task to illustrate shortcomings of current position addressing techniques that are based on token positions. Let us consider simple sequences consisting of two types of tokens $x$ and $y$ to illustrate the interplay of the context and position addressing mechanisms. Given a sequence $yyyyxyyy$, for example, context addressing can focus the attention on token $x$ by producing key and query vectors such that

$$\mathbf{q}^\top \mathbf{k}_x = \mathbf{q}^\top \mathbf{k}_y + \Delta \quad \text{where } \Delta > 0. \tag{2}$$

This will give attention weights $a_x/a_y = \exp \Delta$. Suppose $\Delta = 1$, then the attention on $x$ will be about $e \approx 2.7$ times larger than of $y$. Similarly, position addressing allows us to extract the $i$-th token (in relative position so $i = 0$ is the last token) using position embeddings such that

$$\mathbf{q}^\top \mathbf{e}[i] = \mathbf{q}^\top \mathbf{e}[j] + \delta \quad \text{where } \delta > 0 \text{ and } j \neq i.$$

More interestingly, context and position addressing can work together to do more complex attention such as finding the last $x$ in the sequence $yyxyyxyy$. If we assume $x$ tokens have the same context representation (i.e. the same key vectors), their attention difference will only depend on their positions $i$ and $j$:

$$\frac{a_{x[i]}}{a_{x[j]}} = \exp\left(\mathbf{q}^\top \mathbf{e}[i] - \mathbf{q}^\top \mathbf{e}[j]\right) > \exp(\delta).$$

For the last $x$ at position $i$ to have larger attention, their difference should be larger than some $\delta > 0$. Since the positions $i$ and $j$ are unknown beforehand, the above inequality must hold for any $i < j$, including when $j = i + 1$. Then we can derive

$$\mathbf{q}^\top \mathbf{e}[0] - \mathbf{q}^\top \mathbf{e}[i] > i\delta \quad \text{for } 0 < i.$$

Now let us use $\Delta$ from Eq. (2) and compare to the attention on $y$ at position 0.

$$\frac{a_{x[i]}}{a_{y[0]}} = \exp\left(\mathbf{q}^\top \mathbf{k}_x + \mathbf{q}^\top \mathbf{e}[i] - \mathbf{q}^\top k_y - \mathbf{q}^\top \mathbf{e}[0]\right) < \exp\left(\Delta - i\delta\right)$$

From this, we can see that $y$ will have larger attention than $x$ when $i > \Delta/\delta$, thus the model cannot attend to the last $x$ if it is too far away. This gives us an intuition why independent position and context addressing might fail on very simple tasks.

### 3.2 STATE-OF-THE-ART LLMS FAIL ON COUNTING PROBLEMS

Basic failures of standard position encodings can be observed even in state-of-the-art LLMs. In Table 1, we show a simple word counting task that should be trivial for capable LLMs. Surprisingly, both GPT4 and Llama-2 70B Chat fail on this task. What makes this task challenging for PE is that the model needs to attend to the last sentence while ignoring the one before. The number of tokens in a sentence varies greatly, making token position imprecise. However, if positions were measured in terms of number of sentences instead of tokens, we argue that this task is easy as the model will then attend correctly. See Appendix A for more details on this experiment.

Table 1: Even powerful LLMs struggle to attend to abstract elements like sentences by their position. In this example, both the words "Alice" and "book" are mentioned in the first sentence, not the last. Addressing by token position is not very useful in this case because we do not know how many tokens the last sentence has. Encoding sentence position could make this task trivial.

---

**Prompt:** Alice was beginning to get very tired of sitting by her sister on the bank, and of having nothing to do: once or twice she had peeped into the book her sister was reading, but it had no pictures or conversations in it, "and what is the use of a book," thought Alice "without pictures or conversations?"

So she was considering in her own mind (as well as she could, for the hot day made her feel very sleepy and stupid), whether the pleasure of making a daisy-chain would be worth the trouble of getting up and picking the daisies, when suddenly a White Rabbit with pink eyes ran close by her.

Now, tell me how many times word "Alice" is mentioned in the last sentence.

---

**GPT4:** The word "Alice" is mentioned 1 time in the last sentence.

---

**Llama-2 70B Chat:** The word "Alice" is mentioned twice in the last sentence ...

---

**Prompt:** [THE SAME TWO SENTENCES]

Now, tell me how many times word "book" is mentioned in the last sentence.

---

**GPT4:** The word "book" is mentioned one time in the last sentence.

---

**Llama-2 70B Chat:** The word "book" is mentioned twice in the last sentence: ...

---

# 4 CONTEXTUAL POSITION ENCODING

In CoPE, positions are measured in a context dependent way rather than being a simple token count. The method works by first deciding which tokens should be included when measuring distance using their context vectors. To do that, a gate value is computed for every query $\mathbf{q}_i$ and key $\mathbf{k}_j$ pair

$$g_{ij} = \sigma(\mathbf{q}_i^\top \mathbf{k}_j), \tag{3}$$

where $j < i$ and $\sigma$ is the sigmoid function. A gate value of 1 means that the key will be counted in the position measurement, while 0 means it will be ignored. For example, to count the sentences between tokens $i$ and $j$, the gate value should be 1 for only sentence separation tokens such as ".". The gates also condition on the query, so each query can have different position measurements if needed. The soft gating function allows differentiation so that the system can be trained with backpropagation.

Next, we compute position values by adding the gate values between the current and the target token

$$p_{ij} = \sum_{k=j}^{i} g_{ik}. \tag{4}$$

Note that if the gates are always 1, then $p_{ij} = i - j + 1$ and we recover token-based relative positions. Thus CoPE can be viewed as a generalization of relative PE. In general, however, $p_{ij}$ can be the count of specific words or word types like nouns or numbers, the number of sentences, or other concepts the Transformer deems useful during training.

Unlike token positions, our position values $p_{ij}$ are not restricted to integers and can take fractional values due to the sigmoid function. This means we cannot use an embedding layer to convert a position value to a vector like in the relative PE. Instead, we use interpolation between integer values. First, we assign a learnable embedding vector $\mathbf{e}[p]$ to each integer position $p \in [0, T]$. Then the embedding for position $p_{ij}$ will be a simple interpolation of the two closest integer embeddings

$$\mathbf{e}[p_{ij}] = (p_{ij} - \lfloor p_{ij} \rfloor)\mathbf{e}[\lceil p_{ij} \rceil] + (1 - p_{ij} + \lfloor p_{ij} \rfloor)\mathbf{e}[\lfloor p_{ij} \rfloor]. \tag{5}$$

Finally, we can compute the attention weights similar to Eq. (1)

$$a_{ij} = \text{Softmax}(\mathbf{q}_i^\top (\mathbf{k}_j + \mathbf{e}[p_{ij}])). \tag{6}$$

In practice, however, computing and storing vectors $\mathbf{e}[p_{ij}]$ uses extra compute and memory. We can make this more efficient by first computing the $\mathbf{q}_i^\top \mathbf{e}[p]$ multiplications for all the integer positions $p$,

and then interpolating the resulting values:

$$z_i[p] = \mathbf{q}_i^\top \mathbf{e}[p] \quad \text{for } p \in [0, 1, \dots, T] \tag{7}$$

$$z_i[p_{ij}] = (p_{ij} - \lfloor p_{ij} \rfloor) z_i[\lceil p_{ij} \rceil] + (1 - p_{ij} + \lfloor p_{ij} \rfloor) z_i[\lfloor p_{ij} \rfloor] \tag{8}$$

$$a_{ij} = \text{Softmax}(\mathbf{q}_i^\top \mathbf{k}_j + z_i[p_{ij}]). \tag{9}$$

See Appendix B for more practical implementation details of CoPE.

**Limited positions** From Eq. (4), we can see the maximum value for $p_{ij}$ is the context size $T$, which means we need $T + 1$ position embeddings (including position 0). However, if the gates are sparsely activated (e.g. counting sentences), we can cover the whole context $T$ with much fewer positions. Thus we can set a limit $p_{max} < T$ on the maximum possible position by setting $p_{ij} \leftarrow \min(p_{ij}, p_{max})$.

**Multi-head attention** So far, CoPE is defined for single-head attention. The multi-head extension is straightforward as each head will do their own CoPE independently. The keys and query vectors are different between heads, so that means they can implement different position measurements. For example, head 1 can have keys that turn all gates *on* so that the position counts tokens, while head 2 gates are *on* only for word-beginning tokens, to count words as positions. While the position embeddings $\mathbf{e}[p]$ are shared between the heads only, we also experiment with position embeddings that are shared across the layers as well.

**Computation** The most computationally expensive operation in the self-attention module is the key (or value) and query multiplication that has $\mathcal{O}(T^2 d_h)$ FLOPS, where $d_h$ is the head dimension. The most expensive operation of CoPE is the gate computation in Eq. (3), but we can benefit from the query and key multiplication that was already computed during attention, and reduce gate computation to simply applying the softmax function. The next most expensive operation in CoPE is the matrix multiplication in Eq. (7) that has $\mathcal{O}(T p_{max} d_h)$ FLOPS. This computation can be reduced by selecting a small $p_{max}$, which we show works well in our experiments.

**Computing gates** Note that the same keys are used in computing the gates in Eq. (3) as the final attention computation of Eq. (9). This biases highly attended tokens to be counted in the position computation as well. To disentangle position from attention itself, we can use separate keys that are computed with an additional projection $k_i = W_g h_i$ when computing gates. We denote this version as *sep-keys* in our experiments. Another option is to use the value vectors instead so that $g_{ij} = \sigma(\mathbf{q}_i^\top \mathbf{v}_j)$, which we refer to as *val-gates*. However, these versions will require more computation as we cannot reuse the key query multiplication.

## 5 EXPERIMENTS

In this section we summarize our experimental results. All models were trained and tested on 1 node with 8 GPUs, except the Language and Code Modeling tasks that were trained on 4 nodes (32 GPUs). To further scale up to 1.4B parameters we trained models on 32 nodes (512 GPUs).

### 5.1 FLIP-FLOP TASK

The Flip-Flop language modeling task was introduced in Liu et al. (2024) to expose the failure of Transformer models to capture robust reasoning over long-range input sequences. The input strings consist of alternating sequences of instructions $\{w, i, r\}$ ("write", "ignore", and "read"), each followed by one bit of information (0 or 1) that the model needs to memorize if it follows $w$, or recall the last memory if it follows $r$. It is guaranteed that all strings start with $w$ and end with $r$. For example, given string "$w0i1r0w1i0i1i1r$", the expected output is 1, since the last $w$ operation is followed by 1. To solve this task, the model has to be able to sharply attend to the latest occurrence of the $w$ symbol, the position of which varies between sequences due to ignore instructions. The task defines two test sets: in-distribution and out-of-distribution (OOD), where the latter increases the distance to the last $w$ by increasing the number of ignore instructions.

We replicate the setup described in Liu et al. (2024), and report test error after 10K training steps for models with dimension 256, 4 heads and 4 layers. The results are provided in Table 2 (left). They

Table 2: **Flip-Flop and Selective Copy tasks.** We report in-distribution and out-of-distribution (OOD) generalization test error (%) on both tasks.

| Flip-Flop | | | | Selective Copy | | | |
|---|---|---|---|---|---|---|---|
| PE Method | In-dist | OOD | | PE Method | In-dist | OOD dense | OOD sparse |
| Absolute PE | 6.8 | 21.7 | | Absolute PE | 16.9 | 25.6 | 85.2 |
| RoPE | 1.8 | 20.3 | | RoPE | 40.1 | 100.0 | 100.0 |
| CoPE | **0.0** | **4.9** | | CoPE | **0.0** | **0.0** | **0.0** |

Table 3: **Symbolic counting task** test error rates (%) for different number of variables.

| Symbolic Counting | | | |
|---|---|---|---|
| PE method | 1 var | 3 var | 5 var |
| Absolute PE | 5.3 | 67.6 | 71.5 |
| Relative PE | 1.1 | 17.8 | 22.4 |
| CoPE | **0.0** | **1.2** | **7.4** |

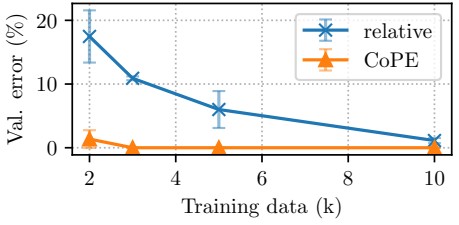

Figure 2: CoPE outperforms relative PE on the symbolic counting, especially with less training data of the task.

show that CoPE outperforms existing methods, allowing the model to not only learn the in-distribution task, but also to generalize to OOD sequences — a property that existing PE methods fail to provide. This is possible because CoPE allows the model to attend to the last seen positions of specific tokens by incorporating their counts into the positional embedding using their keys, i.e. by making the gating function switch on for those tokens. For example, if the gates are 1 only on $w$ tokens, then position 1 will correspond to the last $w$ instruction. In contrast, relative PE struggles to isolate the last $w$ as shown in Section 3.1, especially when its position is unknown and far away.

We also investigate the robustness of the model varying the model dimension, number of heads and layers, with full results reported, including standard deviations, in Appendix C. We find that CoPE is generally robust to these changes with respect to in-distribution generalization, but out-of-distribution generalization can degrade on this task for certain hyperparameter choices.

## 5.2 SELECTIVE COPY TASK

The selective copy task introduced by Gu and Dao (2023) requires context-aware reasoning for selective memorization. In this task the model is given a sequence of tokens and asked to copy all tokens except a denoted blank token. For example, when the input is *DBBCFBFBE* where *B* is the blank, the model is expected to output *DCFFE*. In our experiments, we set the vocabulary size to 16, and the output sequence length (number of non-blanks) to 256, and vary the number of blank tokens. The training and in-distribution test data have 256 blanks whereas the dense and sparse OOD test data have 128 blanks and 512 blanks, respectively. We train models with dimension 64, 2 layers and 2 heads, and report test error after 100k steps. The results, given in Table 2 (right), show that on the in-distribution test set our method CoPE can solve the task while others fail to do so. Similarly, CoPE generalizes better on both dense and sparse OOD test sets. The presence of blank tokens makes it harder to locate the next token to copy, but CoPE can count only non-blank tokens, and hence be more resilient to blanks. At each step, it can then simply copy the non-blank token a distance of 256 (non-blanks) away. Repeating this 256 times will copy the entire sequence of non-blanks.

## 5.3 SYMBOLIC COUNTING TASK

Counting things is more challenging than simply recalling the last instance because it requires more uniform attention over a certain span. For example, to count verbs in the current paragraph, the model needs to attend to the verb tokens roughly equally within the current paragraph. Thus, simple recency bias using position embeddings will not work because it will suppress verbs that occur earlier.

Table 4: **Out-of-distribution (OOD) generalization error (%) on the symbolic counting task**. We vary weight $w_{pass}$ of the dummy `pass` command so the context is either shorter or longer. CoPE generalizes better as it learns to exclude irrelevant `pass` commands from position measurement.

| PE method | in-domain ($w_{\text{pass}} = 50$) | OOD longer context ($w_{\text{pass}} = 100$) | OOD shorter context ($w_{\text{pass}} = 10$) |
|---|---|---|---|
| Relative PE | 1.1 | 8.8 | 34.1 |
| CoPE | **0.0** | **0.0** | **4.0** |

Table 5: Evaluation on the WORDCOUNT task (finetuned 1.4B parameter models).

| Position embedding | WORDCOUNT | | WORDCOUNT-HARD | |
|---|---|---|---|---|
| | NLL | Accuracy | NLL | Accuracy |
| RoPE | 0.055 | 94.8% | 0.24 | 66.9% |
| CoPE + RoPE | **0.045** | **96.0%** | **0.18** | **77.7%** |

To demonstrate this in a controlled setting, we devise a simple algorithmic task that requires counting. The context is a sequence of operations of three types: set variable to zero, increment it, and do nothing. Here is an example "`...; pass; pass; a = 0 ; pass; a ++; pass; pass; a ++; print a 2`". At the end of each sequence there is a print operation that outputs the current value of that variable. This is a fairly simple task as the model just needs to count `++` operations since the last set operation. In a more challenging version of this task, we mix multiple variables in a single sequence.

Similar to the Flip-Flop task, we randomly select one from the 3 types of operation according to the predefined weights $w_{\text{set}} = 1$, $w_{\text{incr}} = 7$, and $w_{\text{pass}} = 50$. We limit the maximum numerical value to be 10. To test OOD generalization, we modify $w_{\text{pass}}$ so that the average length of the relevant context (from the last set operation to the current step) is either longer or shorter. We generate 10K sequences for training, each containing up to 512 operations. We report the average of 3 random seeds.

Results are given in Table 3 and Fig. 2. The baseline model with relative PE struggles to learn this task, especially when there is more than one variable to track. Absolute PE performs even worse. The best performance comes from CoPE, with a perfect score for the 1 variable case. For OOD generalization, relative PE shows poor generalization, while CoPE generalizes very well as shown in Table 4. See Appendix Table 10 for standard deviations of these experiments.

### 5.4 WORDCOUNT TASK

So far, we have only conducted experiments with symbolic tasks. To understand if the proposed mechanism works in a similar way using natural language, we consider the counting problems described in Section 3.2, where we experiment with introducing a new WORDCOUNT task. In this task, the model is asked to count how many times a specific word occurs in the last $k$ sentences. To run experiments, we used the TINYSTORIES dataset (Eldan and Li, 2023) to generate 2.1M and 21k stories for training and validation. To make the task more challenging, we randomly concatenate three different stories to form a sample, thus increasing context size three-fold, forming WORDCOUNT-HARD task. We report the results of finetuning 1.4B model on 1.5B tokens of WORDCOUNT task (training and evaluation details can be found in Appendix E). Our results summarized in Table 5 demonstrate that CoPE significantly improves over standard embedding approaches, boosting the model's capabilities on a complex context-depended task. To see how difficult this task is, we tested powerful Llama-3.1-70B model, which gave only 3.4% accuracy score on WORDCOUNT-HARD.

### 5.5 LANGUAGE MODELING

Next, to test our method on realistic natural language tasks, we conduct experiment with language modeling. We use the Wikitext-103 dataset (Merity et al., 2017), which consists of 100M tokens extracted from Wikipedia. We train a Transformer model that matches the architecture of GPT-2 (Radford et al., 2019) with 12 layers and a hidden size of 768. We train with the negative log-

Table 6: **Wikitext-103 and Code results.**

| Wikitext-103 | | | | Code | | |
|---|---|---|---|---|---|---|
| PE Method | Params (M) | Val. PPL | Test PPL | PE Method | Params (M) | Test PPL |
| Absolute PE | 124.4 | 23.96 | 24.87 | Absolute PE | 20.8 | 4.7 |
| Relative PE | 123.7 | 22.90 | 23.81 | RoPE | 19.8 | 4.1 |
| CoPE | 123.7 | 22.55 | 23.46 | CoPE | 20.8 | **3.9** |
| CoPE + Relative | 123.7 | **22.31** | **23.23** | CoPE + RoPE | 20.8 | 4.0 |

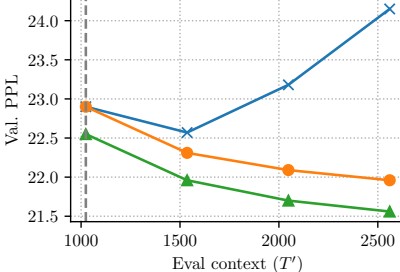 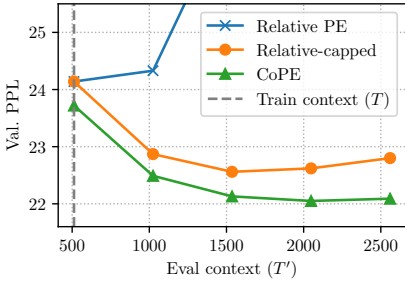

Figure 3: **Generalization to longer context length.** After training on the Wikitext-103 language modeling task with a context size of 1024 (left) and 512 (right), we evaluate the model on longer context sizes and report the validation perplexity. CoPE generalizes well, outperforming existing PE methods, especially when evaluation context size is much larger than training context size (right).

likelihood loss for 100 epochs using a batch size of 64. The model has a context size of 1024, but we set the maximum position value in CoPE to a much lower value of $p_{\max} = 64$.

We compare different PE methods in Table 6 (left). Absolute PE performs worst. CoPE outperforms relative PE, and improves even further when combined with relative PE. This shows that even in general language modeling, CoPE brings improvement.

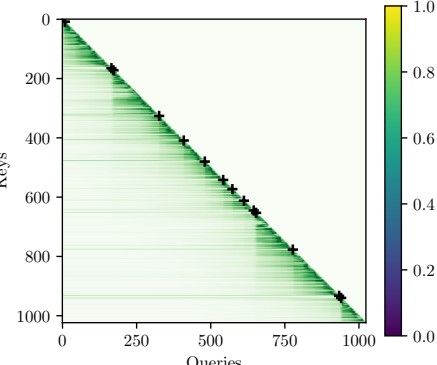 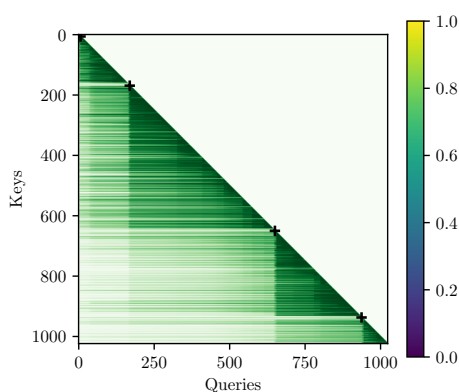

Figure 4: **CoPE can focus attention on abstract elements like current paragraph (left) and section (right)**. Here we show attention induced by position alone on Wikitext-103. Since CoPE is contextualized, it can attend to paragraphs and sections by their position. On the left, the segments are found to be separated by newline tokens (indicated by black plus signs), while the right is separated by section titles like "= = Description = =" (similarly marked).

**Generalization to longer context:** Next, we test how well CoPE generalizes to contexts longer than it was trained on. As CoPE assigns positions conditioning on context, it is capable of distributing them to a much larger number of tokens. While the number of tokens was fixed during training, the number of positions will vary depending on each sample. Thus it is possible that tokens outside the training span of $T$ still get position values that are within the maximum limit $p_{\max}$.

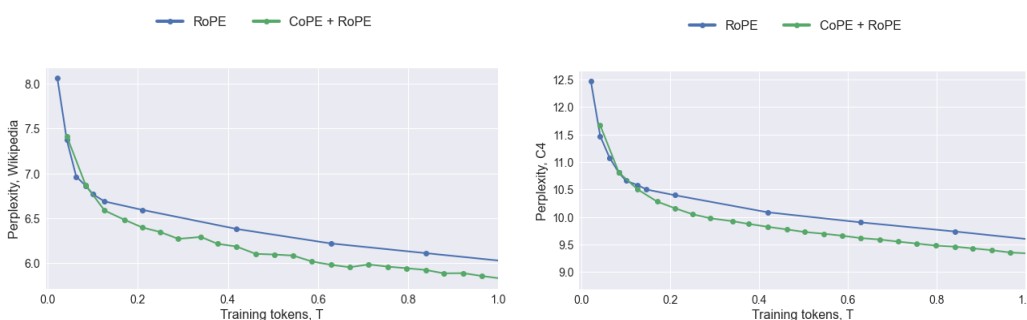

Figure 5: Validation perplexity on Wikipedia and C4 (Raffel et al., 2020) datasets. The model with CoPE embeddings initially lags behind, but after about 10% of training catches up and outperforms the model trained with only RoPE embeddings.

In contrast, relative PE has embeddings that are tied to each token position. Therefore when there are $T' - T$ unseen positions during test time, those tokens will have no position embedding added to them. As this is never seen during training, it negatively affects the performance. To mitigate this, we test a version of relative PE where unseen positions use the embedding of the $T$-th position, which might indicate a "far away" position. This is similar to CoPE where positions are capped by a specified limit. We call this version *relative-capped*.

The results are given in Fig. 3. Relative PE generalizes poorly to longer context sizes. The relative-capped version, in contrast, shows much healthier performance. However, CoPE still outperforms it, and the gap widens when the test context is much longer than the training context (see Fig. 3, right).

In Fig. 4, we show examples of attention maps from a model trained with *sep-keys* (gates are computed with separated keys, see Section 4). The attention maps are built from position alone (they have to be multiplied by context attention for the final attention), which gives us better insight into what CoPE is doing. We also normalize so that the maximum attention weight is always 1 for each query. First, we can see that positions are clearly contextualized as the attention tends to drop at specific tokens regardless of their relative positions. A closer look at those tokens reveals that the attentions are mostly focused on the last paragraph (left) or section (right). For clarity, the actual paragraph and section boundaries are marked by black plus signs. In CoPE, this is possible because one attention head can count paragraphs while another counts sections, and then it can focus on position 0 only. For more details, see the gate values shown in Appendix Fig. 7, and further ablation results in Appendix D.

### 5.6 CODE MODELING

We further test the ability of CoPE by evaluating on code data. Code data has more structure compared to natural language, and might be more sensitive to in-context learning. We train a small 20M Transformer model that resembles the Llama-2 architecture with the corresponding mix of code data (Touvron et al., 2023b) with 4 layers, 8 heads, and a hidden dimension of 256. We use context length 4096, learning rate $5.0e - 4$, and train for 100B tokens.

The results are summarized in Table 6 (right). CoPE embeddings improve in perplexity over absolute PE and RoPE by 17% and 5% correspondingly. Combining RoPE and CoPE embeddings together improves over RoPE, but does not bring any improvements over the proposed embedding method.

### 5.7 LARGE LANGUAGE MODELING (PRE-TRAINING)

Next we test our method on a realistic setup where we scale the language models described in the previous section up to 1.4B parameters by increasing the number of layers to 24, the number of heads to 16, and the hidden dimension to 2048. We then conduct experiments of full scale pre-training using our CoPE architecture compared to the standard RoPE architecture.

The training data and hyperparameter setup mostly repeats the one from Touvron et al. (2023b), but we increase the learning rate to $4 \times 10^{-4}$. To speed up training and minimise the additional number

of parameters, we have added CoPE to every 6th layer of the 24-layer model only (i.e. in total on 4 layers), and limited the maximum position $T$ to 64. Each head (CoPE embedding) dimension is 2048, so the total number of added parameters constitutes only $0.04\%$ from the total. This setup allowed us to achieve on-par runtime during training. We trained on a total of 1T tokens.

Results are given in Figure 5. We observe around $3\%$ improvement in validation perplexity for our model trained with CoPE encodings compared to the RoPE baseline. This leads to much faster convergence, matching the baseline performance 40% faster (0.6T vs 1.0T) when evaluated on Wikipedia. We also perform evaluation on several standard few-shot benchmarks, described in detail in Appendix E, where we also see gains on 6 out of 9 benchmarks.

## 6 RELATED WORK

While the attention mechanism was proposed in Bahdanau et al. (2014) for processing sequences of tokens, the model was still based on RNNs so position encoding (PE) was not necessary. The Memory Network (Weston et al., 2015) architecture moved away from RNNs when processing sequences, instead using multiple layers of attention, and first introduced using PE together with the attention mechanism (Sukhbaatar et al., 2015). They added learnable embedding vectors that correspond to each relative position to the hidden representations. A similar position embedding was used earlier in a convolution-based architecture (Collobert and Weston, 2008), and later in an architecture that combines convolution with attention (Gehring et al., 2017). The latter used an absolute PE because relative position cannot be defined on the source text in machine translation.

PE became in an important topic of research with the popularity of the Transformer architecture. The original paper by Vaswani et al. (2017) employed an absolute PE with fixed vectors, but the relative position embedding was later used in Shaw et al. (2018). Relative PE is especially suited to processing unbounded sequences (Dai et al., 2019). Since then, many different variations of relative and absolute PE have been proposed. In Raffel et al. (2020), each relative position is assigned a simple bias scalar that gets added to the attention logits. While being efficient, this makes position addressing independent of the current token. Press et al. (2022) further simplifies the bias terms by making them fixed in a decaying pattern instead of learning for generalization to longer context. Haviv et al. (2022) takes it to the extreme by removing PE and demonstrated that position information can be recovered by counting previous tokens with causal attention.

While absolute PE was used in early LLMs (Radford et al., 2019), relative PE is more common in recent LLMs (Touvron et al., 2023b;a; Jiang et al., 2023). In particular, RoPE (Su et al., 2024) made it possible to do relative PE without modifying the self-attention code. It relies on the fact that query and key dot product only depend on the angle between those vectors and are agnostic to their absolute angles. Thus if they are rotated by angles proportional to their absolute position, then its effect on the attention logit will only depend on their difference in position. However, CoPE differs from all these PE methods as it measures position in a context dependent way instead of simply using token counts.

While RNNs can be inserted into the Transformer architecture to represent position information in an implicit way (Wang et al., 2019; Neishi and Yoshinaga, 2019), the sequential nature of RNN operations breaks the parallelization of Transformer training, making it slower and less practical. In comparison, the only sequential operation in CoPE is a cumulative sum, which is lightweight and can be partially parallelized. For more details on different PE methods, see the survey by Dufter et al. (2022). Zhao et al. (2023) also provides a survey focused on length generalization of PE methods.

## 7 CONCLUSION

In this paper, we proposed a novel position encoding method called CoPE that measures position in a context dependent way, thus moving away from the current token-based position paradigm. This approach allows more freedom when addressing by position, and brings gains on several tasks. While this paper only focused on text and code domains, CoPE has the potential to improve domains such as video and speech where token position seems intuitively even less appropriate. Another avenue to explore is training even larger models with CoPE beyond the 1.4B parameter models in this paper, which requires considerable pre-training compute budget.

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

## A  BASIC FAILURES OF STANDARD POSITION ENCODINGS IN STATE-OF-THE-ART LLMs

Basic failures of standard position encodings can be observed even in state-of-the-art LLMs. In Table 7, we show detailed prompts for a simple word counting task that should be trivial for capable LLMs. Surprisingly, both GPT4 and Llama-2 70B Chat fail on this task. What makes this task challenging for PE is that the model needs to attend to the last sentence while ignoring the one before. The number of tokens in a sentence varies greatly, making token position imprecise. However, if positions were measured in terms of number of sentences instead of tokens, we argue that this task is easy as the model will then attend correctly. In some cases, we ask a follow-up question to make sure the model is not misunderstood the question.

We accessed GPT-4 through its Playground web interface[2] and used the default sampling parameters of temp= 1 and top-P= 1. For Llama-2 70b Chat we used temp= 0.6 and top-P= 0.9 for sampling.

---

[2]`https://platform.openai.com/playground/chat?models=gpt-4`

Table 7: Full prompts of word counting with powerful LLMs. The follow-up questions makes it clear that indeed LLMs made mistakes.

| | |
|---|---|
| **User** | Alice was beginning to get very tired of sitting by her sister on the bank, and of having nothing to do: once or twice she had peeped into the book her sister was reading, but it had no pictures or conversations in it, "and what is the use of a book," thought Alice "without pictures or conversations?"

So she was considering in her own mind (as well as she could, for the hot day made her feel very sleepy and stupid), whether the pleasure of making a daisy-chain would be worth the trouble of getting up and picking the daisies, when suddenly a White Rabbit with pink eyes ran close by her.

Now, tell me how many times word "Alice" is mentioned in the last sentence. |
| **GPT4** | The word "Alice" is mentioned 1 time in the last sentence. |
| **User** | Where is it mentioned? |
| **GPT4** | The word "Alice" is mentioned in the phrase "thought Alice" in the first paragraph. In the last sentence, however, the word "Alice" is not mentioned at all. |
| **User** | [THE SAME TWO SENTENCES]

Now, tell me how many times word "Alice" is mentioned in the last sentence. |
| **Llama-2 70B Chat** | The word "Alice" is mentioned twice in the last sentence:

1. "So she was considering in her own mind (as well as she could, for the hot day made her feel very sleepy and stupid), whether the pleasure of making a daisy-chain would be worth the trouble of getting up and picking the daisies,"
2. "when suddenly a White Rabbit with pink eyes ran close by her."

Therefore, the answer is 2. |
| **User** | [THE SAME TWO SENTENCES]

Now, tell me how many times word "book" is mentioned in the last sentence. |
| **GPT4** | The word "book" is mentioned one time in the last sentence. |
| **User** | Where is it mentioned? |
| **GPT4** | The word "book" is mentioned in the phrase "the pleasure of making a daisy-chain would be worth the trouble of getting up and picking the daisies, when suddenly a White Rabbit with pink eyes ran close by her." |
| **User** | [THE SAME TWO SENTENCES]

Now, tell me how many times word "book" is mentioned in the last sentence. |
| **Llama-2 70B Chat** | The word "book" is mentioned twice in the last sentence:

1. "the book her sister was reading"
2. "a book without pictures or conversations?"

Therefore, the answer is 2. |

## B  COPE IMPLEMENTATION

```python
class CoPE(nn.Module):
    def __init__(self, npos_max, head_dim):
        super().__init__()
        self.npos_max = npos_max
        self.pos_emb = nn.parameter.Parameter(
            torch.zeros(1, head_dim, npos_max))

```

```python
    def forward(self, query, attn_logits):
        # compute positions
        gates = torch.sigmoid(attn_logits)
        pos = gates.flip(-1).cumsum(dim=-1).flip(-1)
        pos = pos.clamp(max=self.npos_max - 1)
        # interpolate from integer positions
        pos_ceil = pos.ceil().long()
        pos_floor = pos.floor().long()
        logits_int = torch.matmul(query, self.pos_emb)
        logits_ceil = logits_int.gather(-1, pos_ceil)
        logits_floor = logits_int.gather(-1, pos_floor)
        w = pos - pos_floor
        return logits_ceil * w + logits_floor * (1 - w)

class SelfAttn(nn.Module):
    def __init__(self, npos_max, head_dim):
        super().__init__()
        self.cope = CoPE(npos_max, head_dim)
        self.head_dim = head_dim

    def forward(self, query, key, val, mask):
        # q, k, v have dimensions batch x seq_len x head_dim
        attn_logits = torch.bmm(query, key.transpose(-1, -2))
        attn_logits = attn_logits / math.sqrt(self.head_dim)
        attn_logits += mask.log()
        attn_logits += self.cope(query, attn_logits)
        attn = torch.softmax(attn_logits, dim=-1)
        out = torch.bmm(attn, val)
        return out
```

Listing 1: CoPE attention code

## C  FLIP-FLOP EXPERIMENTS

Following Liu et al. (2024), we experiment with Transformer models of different sizes, varying head dimension in $\{128, 256\}$, and number of heads and layers in $\{2, 4\}$. We utilize AdamW optimizer with linear learning rate decay ($lr = 3e - 4, \beta_1 = 0.9, \beta_2 = 0.999, \varepsilon = 10^{-8}$). We train on 8 GPUs with batch size 16 for 10,000 steps. For the main results, we ran 3 seeds and reported their average along with standard deviations as can be seen in Table 8.

In our ablations, we experiment with hard attention, as in this task for each sequence model is required to attend to a single specific token. Furthermore, we experiment with incorporating contextual information into positional encoding through a multilayer perceptron (MLP). In particular, instead of using interpolation (Eq. (5)) we learn the positional encodings by training an $N$-dimensional MLP layer, and denote this approach as CoPE _MLP. This change significantly increases memory and runtime load on the training (by 30-50 times in our experiments compared with regular positional encodings), but allows for more flexibility in positional in-context learning. We vary $N \in \{32, 64, 128, 256\}$, and report results in Table 8 for $N = 64$ to strike the balance between model's efficiency and performance. We also experiment with ingesting CoPE _MLP only in the first layer of the transformer model: this helps to reduce runtime by the order of magnitude, but hurts the performance, especially on the out-of-distribution (OOD) task.

Similarly to the ALIBI approach proposed by Press et al. (2022), we can treat the cumulative sum of the gates as learned biases (while in the original paper authors used static bias). Specifically, Eq. (6) will be simplified to:

$$a_{ij} = \text{Softmax}(\mathbf{q}_i^\top \mathbf{k}_j + m \cdot p_{ij}), \qquad (10)$$

where $m$ is head-specific slope fixed before training. In our experiments on the FlipFlop task, we train model with 4 heads, and experiment with three sets of pre-fixed slopes: $\{1, \frac{1}{2}, \frac{1}{2^2}, \frac{1}{2^3}\}$, $\{\frac{1}{2^2}, \frac{1}{2^3}, \frac{1}{2^4}, \frac{1}{2^5}\}$, $\{\frac{1}{2^4}, \frac{1}{2^5}, \frac{1}{2^6}, \frac{1}{2^7}\}$. We also train a model where $m$ is a learned parameter, specific for each head and layer, and initialized from $0$. No other positional embeddings are added to the model.

Table 8: The test error rates (%) and standard deviation (in parenthesis) on the Flip-Flop task for different Transformer architectures.

| Architecture | Dimension | Number of layers/heads | In-dist. test error | OOD test error |
|---|---|---|---|---|
| Absolute PE | 256 | 4 / 4 | 6.8 (6.9) | 21.7 (7.9) |
| | 256 | 2 / 4 | 11.1 | 10.6 |
| | 256 | 4 / 2 | 0.1 | 18.0 |
| | 256 | 2 / 2 | 13.9 | 31.5 |
| | 128 | 4 / 4 | 5.4 | 24.8 |
| | 128 | 2 / 4 | 0.08 | 19.9 |
| | 128 | 4 / 2 | 0.07 | 16.5 |
| | 128 | 2 / 2 | 19.1 | 28.6 |
| Absolute PE + hard attn | 256 | 4 / 4 | 50.7 | 49.1 |
| RoPE | 256 | 4 / 4 | 1.8 (3.1) | 20.3 (2.9) |
| | 256 | 2 / 4 | 5.1 | 14.7 |
| | 256 | 4 / 2 | 0.02 | 19.0 |
| | 256 | 2 / 2 | 5.4 | 19.8 |
| | 128 | 4 / 4 | 0.1 | 8.9 |
| | 128 | 2 / 4 | 0.1 | 18.2 |
| | 128 | 4 / 2 | 0.02 | 17.3 |
| | 128 | 2 / 2 | 14.4 | 25.2 |
| CoPE | 256 | 4 / 4 | 0.03 (0.06) | 4.9 (4.4) |
| | 256 | 2 / 4 | **0.0** | 13.2 |
| | 256 | 4 / 2 | **0.0** | **3.0** |
| | 256 | 2 / 2 | **0.0** | 14.6 |
| | 128 | 4 / 4 | 0.2 | 33.2 |
| | 128 | 2 / 4 | 0.03 | 22.3 |
| | 128 | 4 / 2 | 0.03 | 14.5 |
| | 128 | 2 / 2 | 0.02 | 24.5 |
| CoPE _MLP | 256 | 4 / 4 | 0.03 | 5.9 |
| CoPE _MLP$_{1st\_layer}$ | 256 | 4 / 4 | 0.9 | 24.3 |
| CoPE _ALiBi ($m[0] = 1$) | 256 | 4 / 4 | **0.0** (0.0) | 11.4 (3.4) |
| CoPE _ALiBi ($m[0] = 1/2^2$) | 256 | 4 / 4 | **0.0** (0.0) | 8.7 (7.6) |
| CoPE _ALiBi ($m[0] = 1/2^4$) | 256 | 4 / 4 | **0.0** (0.0) | 17.1 (1.5) |
| CoPE _ALiBi ($m$ as parameter) | 256 | 4 / 4 | **0.0** (0.0) | 11.4 (4.0) |

We observe higher convergence rate for models with CoPE, reaching lowest in- and out-of-distribution test errors at 2500 steps (Fig. 2). Models with CoPE _MLP also reach near-zero test error rate on in-distribution test set, but require twice as more steps to reach this performance, while transformers with absolute PE fail to learn the task. CoPE _ALIBI-based models show competitive performance, slightly lagging behind on the out-of-distribution task.

## D  ADDITIONAL ABLATIONS

In this section, we summarize the results of our ablation experiments on Wikitext-103 task (see Table 9). We find that computing gates using values (value-gates) instead of keys, or using separate keys (sep-keys) slightly improve perplexity scores on this task. However, these changes come with additional compute, and extra parameters in the case of sep-keys. Next, the position embeddings are only shared among attention heads instead of the whole model, but that does not affect the performance much. Finally, we try decreasing and increasing the number of positions $p_{max}$. We see that even having only $p_{max} = 16$ positions for the context size of $T = 1024$ does not negatively affect the performance, indication that CoPE uses positions more effectively over long range. Finally, we also experiment with ALIBI version of CoPE using Eq. (10) using the recommended slope

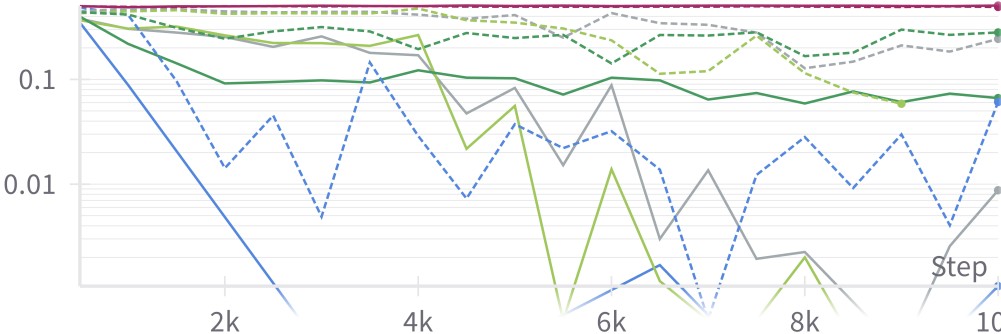

Figure 6: Test error rate on the Flip-Flop task for different Transformer architectures measured every 500 steps. Model with CoPE achieves faster convergence, reaching lowest in- and out-distribution test errors at 2500 steps.

parameters from Press et al. (2022). The performance is worse and roughly matches absolute PE, perhaps because ALɪBɪ slopes are tuned to token positions and lack the flexibility of the position embeddings.

Table 9: Wikitext-103 ablations

| Changes | Params (M) | Val. PPL | Test PPL |
|---|---|---|---|
| None | 123.7 | 22.55 | 23.46 |
| Use val-gates | 123.7 | 22.40 | 23.33 |
| Use sep-keys | 130.8 | **22.39** | **23.18** |
| Layers do not share embeddings | 123.7 | 22.56 | 23.58 |
| $p_{max} = 64 \rightarrow 16$ | 123.7 | 22.45 | 23.22 |
| $p_{max} = 64 \rightarrow T = 1024$ | 123.7 | 22.46 | 23.31 |
| CoPE_ALiBi | 123.7 | 24.16 | 25.09 |

In Table 10 and Table 11 we also report standard deviations on the symbolic counting task and selective copy task.

Table 10: Standard deviation (in parenthesis) of the test error rates on the symbolic counting task

| PE method | 1 var | 3 var | 5 var | $w_{pass} = 50$ | $w_{pass} = 100$ | $w_{pass} = 10$ |
|---|---|---|---|---|---|---|
| Absolute PE | 5.3 (0.8) | 67.6 (1.5) | 71.5 (1.5) | - | - | - |
| Relative PE | 1.1 (0.4) | 17.8 (7.8) | 22.4 (5.1) | 1.1 (0.4) | 8.8 (1.1) | 34.1 (2.5) |
| CoPE | **0.0** (0.0) | **1.2** (2.1) | **7.4** (8.5) | **0.0** (0.0) | **0.0** (0.0) | **4.0** (4.1) |

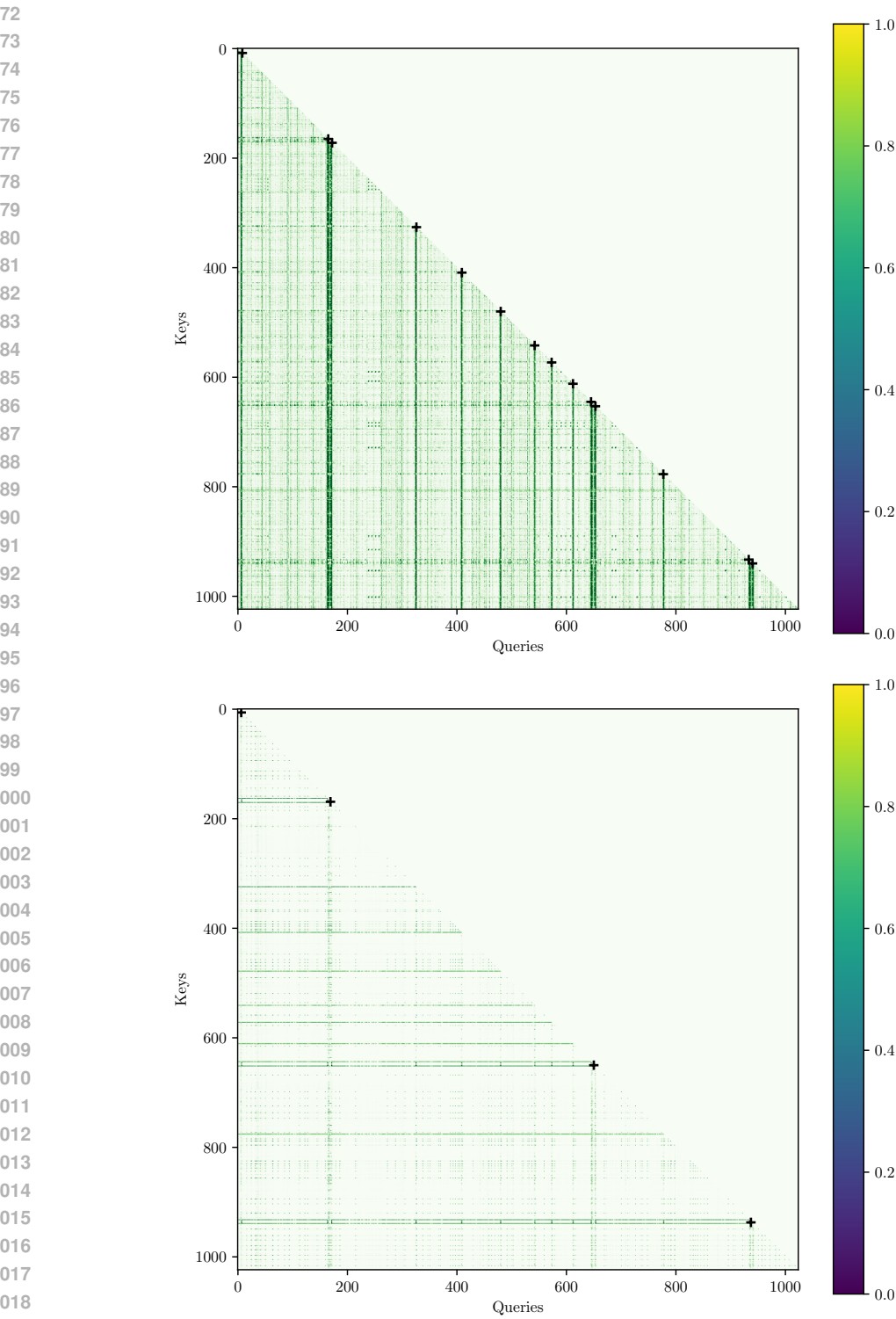

Figure 7: The gate values corresponding to Fig. 4. The gate activations suggest that CoPE is counting paragraphs (top) or sections (bottom).

Table 11: Standard deviation (in parenthesis) of the test error rates on the selective copy task

| PE Method | In-dist | OOD dense | OOD sparse |
|---|---|---|---|
| Absolute PE | 16.9 (3.7) | 25.6(3.8) | 85.2(8.4) |
| RoPE | 40.1(3.5) | 100.0(0.0) | 100.0(0.0) |
| CoPE | **0.0(0.0)** | **0.0(0.0)** | **0.004(0.006)** |

## E  LLM EVALUATION AND FINETUNING DETAILS

First we evaluate our 1.4B model performance on a set of popular benchmarks: BoolQ (Clark et al., 2019), PIQA (Bisk et al., 2020), SIQA (Sap et al., 2019), HellaSwag (Zellers et al., 2019), WinoGrande (Sakaguchi et al., 2021), ARC easy and challenge (Clark et al., 2018), Open-BookQA (Mihaylov et al., 2018). We also report 5-shot performance on the aggregated MMLU benchmark (Hendrycks et al., 2020). Results summarized in Table 12 show that model trained with CoPE embeddings outperforms baseline by $1\%$ on average, even though contextual encodings were added only to $1/6$ of layers and had limited context window.

| Embedding method | BoolQ | PIQA | SIQA | HellaS | WinoG | ARCe | ARCc | OBQA | MMLU | Avg |
|---|---|---|---|---|---|---|---|---|---|---|
| RoPE | 61.4 | 73.0 | 41.2 | **45.5** | 60.4 | 64.3 | **30.2** | 25.2 | **27.8** | 47.7 |
| CoPE + RoPE | **65.8** | **73.4** | **41.7** | 45.4 | **62.0** | **66.2** | 29.7 | **26.8** | 26.6 | **48.6** |

Table 12: 1.4B model performance on standard benchmarks. Training with CoPE+RoPE embeddings outperforms RoPE-based model after training on 1T tokens.

Next, we generate WORDCOUNT and WORDCOUNT-HARD datasets. We use training partition of the TINYSTORIES dataset, and further randomly split it in 100:1 ratio to form train and validation sets. As a target word, we select most common word in the story, and randomly choose number of last sentences to count over $k$ between 1, and total number of sentences in the story. To split story into sentences we use PUNKTSENTENCETOKENIZER from Python's nltk library. We use prompt template displayed below to generate data for training and validation.

> **WORDCOUNT prompt**
>
> [INST] {<story>} How many times word '{<target word>}' mentioned in the last '{<k>}' sentences? [/INST]

We end up with 2,098,294 stories for training, and 21,195 for validation in WORDCOUNT task. To generate WORDCOUNT-HARD task, we concatenate three random stories, thus ending up with 699,431 samples for training, and 7,065 for validation. WORDCOUNT-HARD task is more challenging, as it requires to reason over longer context, attend to more sentences, and cont more tokens, as shown in Fig. 8. Table 13 shows the performance of Llama-3.1 instruction finetuned models on WORDCOUNT and WORDCOUNT-HARD tasks. To evaluate zero-shot performance, we append "Only output the number." to the prompt to avoid unnecessary verbosity.

We finetune 1.4B language model from Section 5.7 in supervised manner. The model is trained with a learning rate of $1 \times 10^{-4}$ and dropout 0.1 with batch size 262,144 tokens on a single 8-GPU node, and keep context length at 4096 tokens. Other parameters are kept the same as during pre-training. During inference we used greedy decoding to generate completions. Evaluation results are summarized in Table 5. While both embedding methods can learn base task, CoPE observe significant improvements for WORDCOUNT-HARD. Investigating how the preformance changed between models as shown in

Table 13: Zero-shot accuracy of Llama-3.1-Instruct models on the WORDCOUNT task.

| Model | WORDCOUNT | WORDCOUNT-HARD |
|---|---|---|
| Llama3.1-8b-Instruct | 17.1% | 8.4% |
| Llama3.1-70b-Instruct | 16.4% | 3.4% |

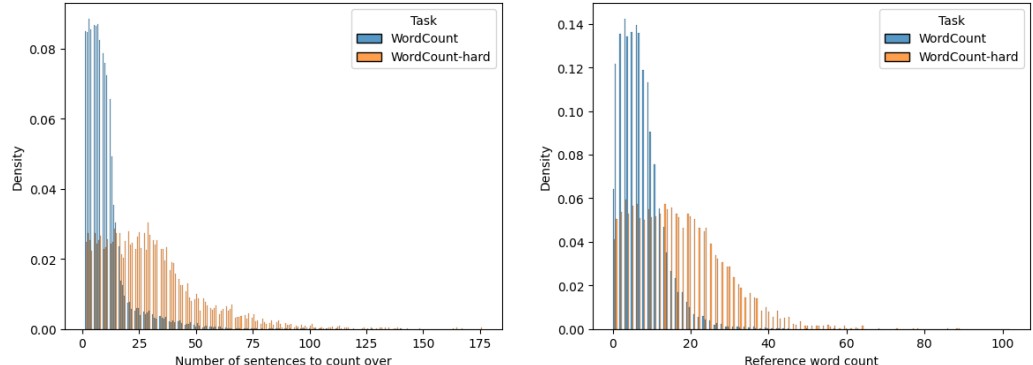

Figure 8: Change in sentence and word distribution between WORDCOUNT and WORDCOUNT-HARD tasks.

Fig. 9, we found that the improvement is drown by long-context task, where the model is asked to count over larger amount of sentences.

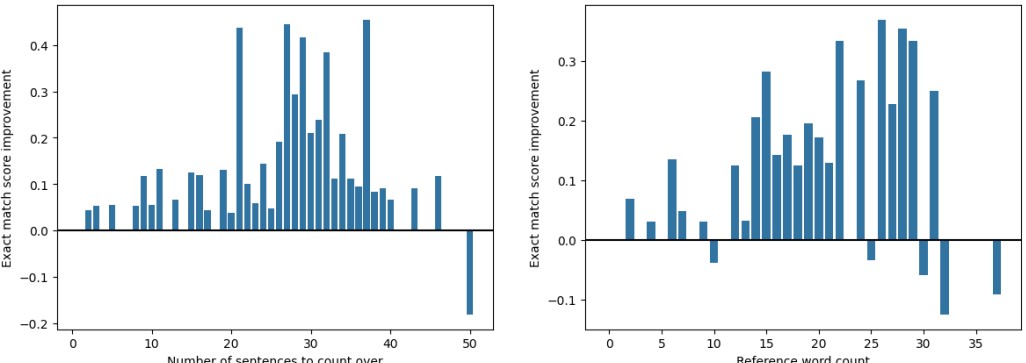

Figure 9: Change in exact match score between models trained with CoPE and RoPE embeddings, calculated over subsets of data with at least 10 datapoints. Models trained with CoPE are better at counting over long distances.

## F    LIMITATIONS

In this paper, we propose a novel position encoding method, that allows positions to be conditioned on context. In our experiments, we mostly focused on tasks where we would expect traditional embedding methods to fail. We also tested our approach on two larger-scale datasets (Wikitext-103 and Code collection), as well as popular benchmarks. However, we did not test how CoPE will perform if scaled beyond 1.4B parameters.

