# OpenReview forum: "Contextual Position Encoding: Learning to Count What’s Important"
_ICLR.cc/2025/Conference — Submitted to ICLR 2025_

### Official Review · Reviewer_cvj8 · 2024-10-31

**Soundness:** 2
**Presentation:** 2
**Contribution:** 2
**Rating:** 5
**Confidence:** 4

**Summary:**

This study presents a position-encoding (PE) method that is context-dependent. The authors use token similarities to calculate a score for each pair of tokens in the sequence. In simple terms, this score represents how many times the query token (or a token very similar to the query token) is seen until now in the sequence. The authors argue that the proposed way of counting allows the capturing of various abstractions in a given sequence, such as token, sentence, paragraph, section etc. With multiple experiments, the authors show the effectiveness of the proposed method CoPE, especially in tasks where counting of words is important.

**Strengths:**

- The study presents a very unique way of modeling positions for the transformer-based language models

- The authors show the advantages of the proposed methods on multiple tasks such as flip-flop, symbolic counting, word counting, language modeling, and common benchmarks.

**Weaknesses:**

- Scalability: The presented method is a parameterized method of encoding positions. The authors mention that to control the number of added parameters one could control the value of p_max (line 227). However, the effect of limiting p_max on performance is unclear, especially for longer sequences (say >10k tokens). Secondly, for CoPE one would need to write ‘qk’ dot product and hence cannot use the sub-quadratic attention methods such as flash attention and ring attention. This is a crucial limiting point because sub-quadratic attention has been a key for scaling transformer LLMs to process longer sequences. Furthermore, if one is not able to use flash/ring-attention for training the model then naturally resorting to shorter sequences is a practical solution. This brings me to the last scalability issue. Interpolation and extrapolation methods have been published in the literature to effectively take an LLM trained on shorter sequences to extend the capabilities to longer sequences. However, how these methods can adapt to parameterized PE methods is a challenging question. The authors show in Section 5.5 (figure 3) that the model trained on a sequence length of 512 can have reducing PPL values until the context reaches 2.5k tokens. It is then critical to see how long this pattern stays stable. Or if the authors could present some analysis regarding the relationship between p_max and extension limit for different model sizes, that’d be helpful for the readers.

- It is also crucial to report cost analysis, including the additional time and memory required by the CoPE method compared to RoPE (with and without flash attn), in order to compare the methods fairly. This will give readers and practitioners a clear idea of the required compute investment for better performance.

- PE for different levels of abstraction: The authors' primary argument concerns a feature of positional encodings (PE): that the PE method should be able to differentiate between a token and a sentence (i.e. different levels of abstraction). In other words, the counting of basic units of the text sequences must happen at different paces for different levels of abstraction e.g., counting should be faster for tokens than the sentences. This, I believe, is indeed one feature of the RoPE. RoPE operates on d//2 subspaces and each subspace counts positions (rotation angle) at a different pace. Hence, we can loosely argue that subspace-1 counts tokens and subspace-2 counts sentences, likewise, in a non-discrete fashion. If this is true and both RoPE and CoPE are doing the differential counting then my main concern is that the difference in the results may primarily be due to the difference in the number of parameters between the two. It would be helpful if the authors could show that this is untrue.

- The details regarding experiments are often missing (see below in the questions). Figure 5 and other figures in the appendix are difficult to read.

**Questions:**

Line 41-42: This position variance increases …: Can authors please cite the reference for the sentence length statistic?

Line 251: Experiments
Which GPUs? How much memory?
What are the p_max values for different experiments? The value seems to be reported only for the language modeling experiments.
How is the test error calculated, non-exact match?

Flip-flop task:
Line 864: Table 8 in Appendix: OOD performance of CoPE across different architectures is worse than RoPE. This is further impacted if CoPE parameters are only added for 1st layer. How do the authors explain this variability in the performance?

Mean (STD):
RoPE: 17.59 (4.99)
CoPE: 17.90 (9.69)

Symbolic counting
Line 324: Table 3/4: Why RoPE is not included in this comparison? Can authors also report the standard deviation values for tables 3 and 4?

Word counting task
Line 367: Is it the full fine-tuning or some lighter fine-tuning like LoRA?
Line 371: The reported performance of Llama-3.1-70B is with or without fine-tuning? If it is without fine-tuning then please report how many shots were used?

Language modeling
Why did the authors choose to train for 100 epochs? For pre-training phase, it is generally 1 epoch or some upsampling of the data (x2-4).
Line 378: Table 6: Why RoPE is not reported for the Wikitext experiment?
Line 378: Table 6: Are embedding parameters included in the reported parameters in both tables?
Line 428: context length extension is conducted on the models pre-trained in section 5.5?
Line 452: The capping idea is similar to the Alibi-style context length extension. Is there any specific reason why Alibi (or PI/Yarn) was not selected for comparison in this experiment?
Line 918: Please consider improving Figure 6. It is very difficult to read.
Line 945: Does Table 9 report non-embedding parameters only? Why changing the p_max value does not have any effect on the number of parameters?

Large language modeling
What was the pre-training context length?

---

> ### Author Response · Authors · 2024-11-15
> **Thank you for your comments**
>
> We thank the reviewer for the extensive and detailed review.
>
> **Scalability: The presented method is a parameterized method of encoding positions. The authors mention that to control the number of added parameters one could control the value of p_max (line 227). However, the effect of limiting p_max on performance is unclear, especially for longer sequences (say >10k tokens).**
>
> We didn’t see a significant performance drop when reducing p_max. You can see those results in Table 9 where we tried p_max values of 16, 64, and 1024. Also, limiting p_max is an optional feature and CoPE will work without this limit, as we have done in some of our experiments. Without p_max limit (i.e. set to max), CoPE still has linear computational complexity, which is much lighter weight than the attention computations that are quadratic.
>
> **Secondly, for CoPE one would need to write ‘qk’ dot product and hence cannot use the sub-quadratic attention methods such as flash attention and ring attention. This is a crucial limiting point because sub-quadratic attention has been a key for scaling transformer LLMs to process longer sequences.**
>
> This is not accurate. Exact attention computation requires a QK dot product. Sub quadratic attention methods always have some sort of approximation. Ring attention is not sub-quadratic and computes QK product exactly as it does not do approximation. It makes computation efficient to run on multiple devices (it is distributed through manageable blocks of the input sequence across multiple devices). Note that every operation in CoPE is linear and only quadratic operation is QK and QV multiplications, which are part of normal attention. If there are approximate methods for calculating these in sub-quadratic time, then surely those can be adapted to be used in CoPE as well.
>
> **The authors show in Section 5.5 (figure 3) that the model trained on a sequence length of 512 can have reducing PPL values until the context reaches 2.5k tokens. It is then critical to see how long this pattern stays stable.**
>
> We demonstrated generalization to 5x longer sequences, which we think provides sufficient evidence of better generalization. Note that when RoPE is extended to longer sequences, it generally requires adjusting base frequency and finetuning on longer sequences. In our experiments, we do not do any finetuning and directly test OOD generalization to longer sequences. This is possible because the same CoPE embeddings appear in different locations in each sample. Thus, when there is a longer sequence, a position embedding that has been seen in shorter sequences can now appear in distant context, allowing the model to generalize and attend to such distances.
>
> **Or if the authors could present some analysis regarding the relationship between p_max and extension limit for different model sizes, that’d be helpful for the readers**
>
> We provided results with different p_max values in Table 9. We also experimented with diverse model sizes ranging from very small to 1.4B size, covering 7 different tasks. While adding even more experiments could be interesting, the main goal of the paper is to demonstrate the need for contextual position, and we feel the current experimental evidence provides sufficient evidence for that.
>
> **It is also crucial to report cost analysis, including the additional time and memory required by the CoPE method compared to RoPE (with and without flash attn), in order to compare the methods fairly. This will give readers and practitioners a clear idea of the required compute investment for better performance.**
>
> As you can see in the implementation in Appendix B, none of the operations in CoPE has more complexity in terms of computation and memory than what is already done in normal attention. As we discussed above, there is a difference when hardware-aware efficient attention implementations are involved as there is none for CoPE yet. However, we showed that even adding CoPE to only some layers brings a significant improvement while managing the extra computation due to lack of efficient implementation. We hope future work will bring efficient attention to CoPE as well.

---

> > ### Author Response · Authors · 2024-11-16
> > **Thank you for your comments (cont)**
> >
> > **This, I believe, is indeed one feature of the RoPE. RoPE operates on d//2 subspaces and each subspace counts positions (rotation angle) at a different pace. Hence, we can loosely argue that subspace-1 counts tokens and subspace-2 counts sentences, likewise, in a non-discrete fashion.**
> >
> > This is not entirely accurate. While RoPE can measure position with many granularities, it cannot precisely count things with varying length, such as sentences, paragraphs, or sections. The main point of our paper is about such context dependency where word count alone is not sufficient. For example, attending to the last sentence might require attending to 5 tokens or 50 tokens. It is impossible to know the exact number without context. This is why RoPE cannot achieve such contextual position measuring. It can attend to every 50 tokens or every 500 tokens, but that cannot depend on context.
> >
> > **Line 41-42: This position variance increases …: Can authors please cite the reference for the sentence length statistic?**
> >
> > This refers to a simple statistical property. As magnitude increases, so does the variance. Thus the variance of paragraph length is larger than that of sentences.
> >
> > **Line 251: Experiments Which GPUs? How much memory? What are the p_max values for different experiments? The value seems to be reported only for the language modeling experiments. How is the test error calculated, non-exact match?**
> >
> > In all experiments we used NVIDIA A100 GPUs with 80GB memory, except 1.4B model training where we used NVIDIA H100. We did not limit p_max (i.e. used full context length) in all experiments unless stated otherwise. Test error is calculated as exact match. We reported test error as it was reported in the original paper that introduced FlipFlop dataset.
> >
> > **Flip-flop task: Line 864: Table 8 in Appendix: OOD performance of CoPE across different architectures is worse than RoPE. This is further impacted if CoPE parameters are only added for 1st layer. How do the authors explain this variability in the performance?**
> >
> > OOD performance of CoPE across different architectures is *better* than RoPE except two reported cases (for models with 128 dim, 2 or 4 layers and 4 heads). We note that all of these experiments - including these two - where deviation is not reported, were run only one time, so we assume it can be noise or an outlier. As for the reference to the first layer, this is a different architecture - CoPE _MLP - described in lines 840-855. About the variability, we are reporting what we observed in the experiments, and it is hard to give concrete explanations for it. In general, tasks like flip-flops require the model to figure out an algorithm, and once that is found the accuracy goes up fast. Thus, they generally have higher variance compared to LM, where the model has to learn many different things, which occurs more gradually.
> >
> > **Symbolic counting Line 324: Table 3/4: Why RoPE is not included in this comparison? Can authors also report the standard deviation values for tables 3 and 4?**
> >
> > In Table 10 and Table 11 we report standard deviations on the symbolic counting task and selective copy task. We had to move it to the appendix due to the page limit. Instead of RoPE we reported Relative PE. We used two different codebases to make sure there is a less chance of bugs. Those codebases have different position embeddings implemented in them, thus our baselines differ in some experiments.
> >
> > **Word counting task Line 367: Is it the full fine-tuning or some lighter fine-tuning like LoRA? Line 371: The reported performance of Llama-3.1-70B is with or without fine-tuning? If it is without fine-tuning then please report how many shots were used?**
> >
> > Thank you for noting, we are referring to the results reported in Table 13. As stated in Table 13, we report Zero-shot accuracy of Llama-3.1-Instruct models. I.e. no further finetuning was done. We will add the reference in the main text.
> >
> >
> > **Language modeling Why did the authors choose to train for 100 epochs? For pre-training phase, it is generally 1 epoch or some upsampling of the data (x2-4). **
> >
> > Wikitext training is not really comparable to pretraining of LLM. For small data like wikitext, it is common practice to train many epochs.
> >
> > **Line 378: Table 6: Why RoPE is not reported for the Wikitext experiment?**
> >
> > We used two different codebases to make sure there is a less chance of bugs. Those codebases have different position embeddings implemented in them, thus our baselines differ in some experiments.
> >
> > **Line 378: Table 6: Are embedding parameters included in the reported parameters in both tables?**
> >
> > Yes, all parameters - including embeddings - are reported. CoPE adds a very small number of parameters relative to the model size.
> >
> > **Line 428: context length extension is conducted on the models pre-trained in section 5.5?**
> >
> > Yes, it is part of Section 5.5, and was performed in the same setup

---

> > > ### Author Response · Authors · 2024-11-16
> > > **Thank you for your comments (contd)**
> > >
> > > **Line 452: The capping idea is similar to the Alibi-style context length extension. Is there any specific reason why Alibi (or PI/Yarn) was not selected for comparison in this experiment?**
> > >
> > > The capping idea is not specific to Alibi, and can be applied to almost any PE method. For relative PE, we can only apply it to the most recent N tokens for example. We have compared to Alibi version of CoPE.
> > >
> > > **Line 918: Please consider improving Figure 6. It is very difficult to read.**
> > >
> > > This figure in Appendix resembles the middle picture in Figure 3 reported in the original paper where the Flip-Flop task was introduced (Bingbin Liu, Jordan Ash, Surbhi Goel, Akshay Krishnamurthy, and Cyril Zhang. Exposing attention glitches with flip-flop language modeling. Advances in Neural Information Processing Systems, 36, 2024.) The idea was to show how it will change with the CoPE-based model. We will work on changing the formatting to make it more pleasant to read.
> > >
> > > **Line 945: Does Table 9 report non-embedding parameters only? Why changing the p_max value does not have any effect on the number of parameters?**
> > >
> > > No, all parameters - including CoPE embeddings - are reported. CoPE adds a very small number of parameters relative to the model size. It appears the same due to rounding.
> > >
> > > **Large language modeling What was the pre-training context length?**
> > >
> > > More details about training setup is in Appendix E. We kept original 4096 tokens in context (for ex, mentioned in line 1069).

---

> > > > ### Comment · Reviewer_cvj8 · 2024-12-03
> > > >
> > > > I am not satisfied with the responses given by the authors on the following points,
> > > >
> > > > - Writing QK product hinders scalability of COPE
> > > > "Exact attention computation requires a QK dot product. Sub quadratic attention methods always have some sort of approximation." My concern is that because QK is being written it is difficult to scale COPE, authors have not addressed it. Flash-attn has become almost a default choice because it offers significantly less memory consumption with exact attn computation without approximation. I do not understand how COPE can be compatible with flash-attn-like methods.
> > > >
> > > > - OOD performance
> > > > Mean (STD): RoPE: 17.59 (4.99) CoPE: 17.90 (9.69), the authors say "OOD performance of CoPE across different architectures is better than RoPE except two reported cases (for models with 128 dim, 2 or 4 layers and 4 heads). We note that all of these experiments - including these two - where deviation is not reported, were run only one time, so we assume it can be noise or an outlier." I cannot assume something as a noise unless it's shown.
> > > >
> > > > - unreported experiments
> > > > The authors say "We used two different codebases to make sure there is a less chance of bugs. Those codebases have different position embeddings implemented in them, thus our baselines differ in some experiments.". Please perform the correct comparisons, and report all values. Having codebase constraints to not report some numbers is not acceptable.
> > > >
> > > > I am keeping my score the same.

---

> > > > > ### Author Response · Authors · 2024-12-03
> > > > >
> > > > > We would like to address the issues you have raised.
> > > > >
> > > > > **Writing QK product hinders scalability of COPE "Exact attention computation requires a QK dot product. Sub quadratic attention methods always have some sort of approximation." My concern is that because QK is being written it is difficult to scale COPE, authors have not addressed it. Flash-attn has become almost a default choice because it offers significantly less memory consumption with exact attn computation without approximation. I do not understand how COPE can be compatible with flash-attn-like methods.**
> > > > >
> > > > > We still do not quite understand what is the issue here. You say that Flash-attn computes attention without approximation. That means it does compute QK multiplication, as attention is softmax(QK). Thus, we can use that QK multiplication computed by Flash-attn to compute CoPE gate values. There is no need to compute QK separately. Similarly, CoPE can benefit from any efficient attention implementation without approximation.
> > > > >
> > > > > **OOD performance Mean (STD): RoPE: 17.59 (4.99) CoPE: 17.90 (9.69), the authors say "OOD performance of CoPE across different architectures is better than RoPE except two reported cases (for models with 128 dim, 2 or 4 layers and 4 heads). We note that all of these experiments - including these two - where deviation is not reported, were run only one time, so we assume it can be noise or an outlier." I cannot assume something as a noise unless it's shown.**
> > > > >
> > > > > We gave our assumption because we were asked for an explanation in the discussion. We didn’t claim it to be true (or write it in the paper) because we don’t have direct evidence to support it. However, in our experience, such a small model trained on such algorithmic tasks tend to have some noise in the results, especially for OOD evaluations. This can be seen where we did actually measure noise e.g. in Tab.8 (CoPE OOD accuracy of 4.9% with std-dev of 4.4%, Absolute PE in-dist accuracy of 6.8% with std-dev of 6.9%, RoPE in-dist accuracy of 1.8% with std-dev 3.1%). In summary, while we do not have direct evidence of noise in all experiments, related experiments showed large noise, which supports our assumption.
> > > > >
> > > > > **unreported experiments The authors say "We used two different codebases to make sure there is a less chance of bugs. Those codebases have different position embeddings implemented in them, thus our baselines differ in some experiments.". Please perform the correct comparisons, and report all values. Having codebase constraints to not report some numbers is not acceptable.**
> > > > >
> > > > > Our comparisons are correctly done to the best of our knowledge. We correctly compared our method against Relative PE on some toy tasks and small LM experiments, where we showed our method outperforms the baseline. In separate experiments, we correctly compared CoPE to RoPE and showed that our method works better. The main point of the paper is to show that measuring position in a context dependent way is beneficial, and both sets of experiments support that claim. Both sets of experiments are done correctly, so they are acceptable evidence on their own. Note that we are not hiding or failing to report experimental results. We are reporting all the results and evidence we have obtained, which spans a wide range of tasks and settings to support our claims.
> > > > >
> > > > > In addition, Relative PE uses more general “learnable” embeddings in measuring relative token positions, thus it is a better baseline in our toy setups. In contrast, RoPE uses fixed rotation embeddings that are more efficient to compute, thus it makes sense to compare to RoPE in our large scale LM experiment, but less so in toy experiments. It is unrealistic to ask us to use Relative PE in our 1.4B scale pre-training given its inefficiency compared to RoPE.

---

### Official Review · Reviewer_4JVS · 2024-11-01

**Soundness:** 2
**Presentation:** 3
**Contribution:** 2
**Rating:** 5
**Confidence:** 4

**Summary:**

This paper proposes a new position encoding method,  CoPE, which enhances the attention mechanism in LLMs by allowing position conditioning based on context, thereby enabling more nuanced interactions such as attending to specific words, nouns, or sentences, and outperforming existing position embeddings in various tasks.

**Strengths:**

1. The paper identifies a novel issue within the PE component of current LLMs and presents insights that could inspire future research directions.
2. The writing is clear, and effective, and conveys the paper's main arguments.

**Weaknesses:**

1. A core issue here is whether the series of problems currently described are truly caused by PE.
2. The analysis in Table 1 attributes the problems to the current PE, but the analysis is insufficient and may be due to the inadequate reasoning abilities of LLMs.
3. One problem with this work is that it requires extensive training; perhaps optimizing during the reasoning phase could resolve these issues (as mentioned in Table 1).
4. Since this work is general, the experiments on generality (Table 12) should not only compare RoPE with RoPE+CoPE but also display the performance of CoPE alone.
5. The content in lines 32 to 36 of the Introduction is somewhat redundant and could be moved to the related work or background section.

**Questions:**

1. Why emphasize OOD in the experiment? For PE, does improved OOD indicate better generalizability and extensibility of the method?
2. Could you provide examples of correct outputs from the experiments in Table 7? This would allow us to better understand the output patterns of LLMs on this task, thereby gaining insights into issues related to PE.

---

> ### Author Response · Authors · 2024-11-15
> **Thank you for your comments**
>
> We appreciate your review of our paper.
>
> **A core issue here is whether the series of problems currently described are truly caused by PE.**
>
> We experimentally demonstrated that changing PE brings improvement in various tasks where position in abstract terms are more useful. This is evidence that PE is a major component of these issues. Position information is crucial in locating N-th sentence or paragraph. But, we argue that position alone is not sufficient, thus proposing a contextualized version. Similar findings in failure mechanisms were demonstrated both empirically and theoretically on context-dependent counting toy tasks in earlier papers, such as [1,3,4], and are main motivators to search for alternative architecture, such as [2].
>
> **The analysis in Table 1 attributes the problems to the current PE, but the analysis is insufficient and may be due to the inadequate reasoning abilities of LLMs.**
>
> Table 1 is meant to be a motivating example highlighting the problem, rather than a quantitative analysis. In Section 5.4, we do careful experimental tests on this word counting issue, and demonstrate that our method helps with it. Given that current LLMs are capable of solving complex math problems, counting words in a sentence is unlikely to be more challenging in terms of reasoning. If only the target sentence is given those LLMs, then they count words easily. This indicates that the problem is inability to isolate and attend to a specific sentence. We do provide extensive analysis of why this might happen in Section 3.1.
>
> **One problem with this work is that it requires extensive training; perhaps optimizing during the reasoning phase could resolve these issues (as mentioned in Table 1).**
>
> We argue that all PE methods require extensive training. To use RoPE, for example, one needs to add it to the model during both pre-training and post-training. The same is true for CoPE. As a part of the architecture, all PE methods usually are trained together with the model through all stages (pre-training and post-training). While we agree that prompt engineering or finetuning might help to steer the model in the right direction, the goal of our approach is to eliminate the core reason of the issues.
>
> **Since this work is general, the experiments on generality (Table 12) should not only compare RoPE with RoPE+CoPE but also display the performance of CoPE alone.**
>
> Please see our response to A1r6 for why we made this comparison. We report results with CoPE alone, as well as CoPE + RoPE, on smaller models where specialized hardware-aware attention implementation is not necessary.
>
> **The content in lines 32 to 36 of the Introduction is somewhat redundant and could be moved to the related work or background section.**
>
> Yes, it can be moved to the background section. We thought some people that are not very familiar with PE may find those parts useful in understanding our motivation.
>
> **Why emphasize OOD in the experiment? For PE, does improved OOD indicate better generalizability and extensibility of the method?**
>
> OOD generalization is generally a good property in most methods as it means the method is robust. We want LLM to perform well in OOD situations because users might ask questions that are different from what training data covered. PE is part of this, so yes PE with better OOD is desirable for building a robust LLM, specifically when the context gets longer or the number of things in the context change.
>
> **Could you provide examples of correct outputs from the experiments in Table 7? This would allow us to better understand the output patterns of LLMs on this task, thereby gaining insights into issues related to PE.**
>
> Table 7 is showing a motivational example showing the shortcoming of LLMs. It is obtained by prompting the LLMs as shown in Table7, with those exact words. The correct output should have had the correct answers, which is that Alice nor book is not mentioned in the last sentence.
>
> References:
>
> 1. Bingbin Liu, Jordan Ash, Surbhi Goel, Akshay Krishnamurthy, and Cyril Zhang. Exposing attention glitches with flip-flop language modeling.
> 2. lbert Gu and Tri Dao. Mamba: Linear-time sequence modeling with selective state spaces.
> 3. Michael Hahn. Theoretical Limitations of Self-Attention in Neural Sequence Models.
> 4. David Chiang and Peter Cholak. Overcoming a Theoretical Limitation of Self-Attention

---

### Official Review · Reviewer_A1r6 · 2024-11-02

**Soundness:** 3
**Presentation:** 3
**Contribution:** 3
**Rating:** 6
**Confidence:** 4

**Summary:**

This paper extends transformers with relative position encodings that can count intervening events. For each query position i and key/value position j, each intervening position k computes a gate value g_{ik} between 0 and 1, and the relative position encoding from i to j depends on the sum of the intervening gate values.

The new encoding, called CoPE, strongly outperforms absolute PEs and RoPE on some synthetic tasks. CoPE+RoPE outperforms RoPE alone on counting words, and improves perplexity on several language modeling tasks.

**Strengths:**

The idea of adding gates and position encodings that count gate values instead of positions is very nice, and seems to be effective at least for the tasks attempted.

**Weaknesses:**

The mapping from a count (p_{ij}) to an embedding (eq. 5) is fine, but the rounding to the nearest integer count seems a little complicated, and learning different embeddings for every integer count adds a lot of parameters and imposes a maximum count. What would be wrong with treating p_{ij} as an angle and converting it to a vector using a sinusoidal encoding? It seems worthwhile to compare this simpler scheme with eq. 5.

Perplexity doesn't seem like the best way to measure language model performance any more; a benchmark like MMLU would have been much more convincing.

**Questions:**

In Table 5, why is only CoPE+RoPE tested? Does CoPE alone not perform well? Even if so, the results should be reported.

---

> ### Author Response · Authors · 2024-11-15
> **Thank you for your comments**
>
> We thank the reviewer for the time and effort.
>
> **The mapping from a count (p_{ij}) to an embedding (eq. 5) is fine, but the rounding to the nearest integer count seems a little complicated, and learning different embeddings for every integer count adds a lot of parameters and imposes a maximum count.**
>
> Our method of interpolating between integer position embedding is simple to implement (see Appendix B), and performs well in practice, as we have shown with our experiments. We have also released our code to ensure reproducibility of our experiments and further exploration. The number of new parameters added is really small and negligible compared to the model size. This can be seen in Table 9 where replacing CoPE embeddings with ALibi bias terms doesn’t bring a noticeable change in the number of parameters. Also Table 6 shows that CoPE does not increase the parameter count compared to RoPE. But we agree that we only explored two methods (interpolation and MLP) of converting fractional positions to embeddings in this work, and future work could find better methods.
>
> **What would be wrong with treating p_{ij} as an angle and converting it to a vector using a sinusoidal encoding? It seems worthwhile to compare this simpler scheme with eq. 5.**
>
> This is a good question. We can treat p_ij as an angle between vectors A and B. However, this is not very practical. A normal RoPE  can be efficiently implemented because A can be a function of i, A=f(i), and B can be a function of j, B=g(j). This makes it RoPE efficient because we only need to add f and g embeddings once per batch. This is not true for CoPE because A and B cannot be expressed by i and j independently. The angle p_ij changes depending on the context and the query. This means there are no A=f(i) and B=g(j) functions exist, making it impossible to implement efficiently.
>
> **Perplexity doesn't seem like the best way to measure language model performance any more; a benchmark like MMLU would have been much more convincing.**
>
> Besides perplexity, we report accuracy metrics in many of our experiments, which is more reliable. The problem with benchmarks like MMLU in our setup is that the average context length is relatively short, thus the benefit of measuring position in a context dependent way is less beneficial. In contrast, when we measure perplexity, the model has a very long context that fills up the entire context window, making it beneficial to attend to far away tokens where abstract positions (e.g. number of paragraphs) are more useful.
>
> **In Table 5, why is only CoPE+RoPE tested? Does CoPE alone not perform well? Even if so, the results should be reported.**
>
> At this scale (1.4B), training using an efficient attention implementation becomes crucial to keep the computational resources at a manageable level. There are specialized very fast attention implementations that exist for RoPE that take advantage of hardware features. We do not have such specialized implementations for CoPE yet, thus we added CoPE to only a few layers while keeping RoPE in all layers to make the pretraining experiment possible. However, we still saw improvements with only a few layers of CoPE, which indicates the usefulness of CoPE.

---

### Official Review · Reviewer_XPkW · 2024-11-08

**Soundness:** 3
**Presentation:** 2
**Contribution:** 2
**Rating:** 5
**Confidence:** 3

**Summary:**

This paper studies word and sentence counting problems for transformer-based language models. The authors observe that transformers require position encodings (PE) to access to a specific word or sentence. They further argue that existing PE methods are context-independent, which leads to limitations in word and sentence counting problems.  Therefore, the authors suggest a new PE method, CoPE, to integrate context and position addressing together. CoPE shares the idea of relative PE but replaces the second term with a learnable vector called contextual position.

To verify the method, the authors conduct experiments on counting problems, long-context problems, language modeling, and code modeling. The experimental results provide evidence to support CoPE.

**Strengths:**

1. Counting is a challenge for models so the research problem is significant.
2. It is interesting to address the problem from the angle of position encodings.
3. Experiments are solid.

**Weaknesses:**

1. The major concern is the novelty. The learnable vector or contextual position is based on the idea of position interpolation and fractional positional encoding explored in Insert Transformer, particularly in [1]
2. Experimental settings are not transparent and consistent.  For example, Table 2 considers Absolute PE and RoPE as baselines, while Tables 3,4 and 5 consider other baselines. I do not see any particular consideration in changing the baseline.
3. There is one unclear point. The authors add CoPE to the model for several layers, while baseline methods are only considered for the input. Is this a factor to affect the final results?
4. In Table 7, LLMs can be prompted to explain counting. It is interesting to show a similar result from CoPE. Therefore, we might understand that LLMs with CoPE can "understand" counting tasks instead of guessing a good number.
5. Minor points for presentations:
    - Word count and word index are not exchangeable in some contexts. It is better to make this more concise. For example, in the abstract "However, current PE methods use token counts to derive position," I suppose this should be "token index" instead of "token count"
    - Visualizations should be consistent. The style of Figure 5 is different from others.



[1] Zhisong Zhang, Yizhe Zhang, and Bill Dolan. 2023. Towards More Efficient Insertion Transformer with Fractional Positional Encoding. In Proceedings of the 17th Conference of the European Chapter of the Association for Computational Linguistics, pages 1564–1572, Dubrovnik, Croatia. Association for Computational Linguistics.

**Questions:**

1. Refers to Weakness.
2. Have you tried your pretrained model for language understanding task?

---

> ### Author Response · Authors · 2024-11-15
> **Thank you for your comments**
>
> Thank you for taking the time to review our paper, but we need to clarify several things:
>
> **The major concern is the novelty. The learnable vector or contextual position is based on the idea of position interpolation and fractional positional encoding explored in Insert Transformer, particularly in [1]**
>
> We respectively disagree. The cited work is significantly different from our work. Our main goal is to measure position in a context dependent way. Note that this problem is not specific to word or sentence counting, but it is well highlighted in the counting tasks we used to demonstrate the new capabilities. In [1] position only depends on token ordering and where tokens are inserted, without any context dependency. It has a completely different goal of computing position embeddings efficiently when tokens are inserted between existing tokens. They achieve this by passing the neighboring position embeddings through a neural network to obtain a new position embedding for the inserted token. This is a completely different mechanism than CoPE where we count positions by adding gated values, then convert them to vectors via interpolation. Thus we argue that the Insert Transformer method won’t be able to resolve failures demonstrated in our paper.
>
> **Experimental settings are not transparent and consistent. For example, Table 2 considers Absolute PE and RoPE as baselines, while Tables 3,4 and 5 consider other baselines. I do not see any particular consideration in changing the baseline.**
>
> We used two different codebases to make sure there is a less chance of bugs. Those codebases have different position embeddings implemented in them, thus our baselines differ in some experiments. There was no intent to be less transparent.
>
> **The authors add CoPE to the model for several layers, while baseline methods are only considered for the input. Is this a factor to affect the final results?**
>
> This is not true. All relative position encoding methods including RoPE have to be added at each layer, and that’s how they are commonly implemented. The same is true for our implementations where those baselines are added at each layer.
>
> **In Table 7, LLMs can be prompted to explain counting. It is interesting to show a similar result from CoPE. Therefore, we might understand that LLMs with CoPE can "understand" counting tasks instead of guessing a good number.**
>
> Yes, that would be an interesting experiment. But such high-level instruction following capabilities emerge only after a very large scale pre-training (not to mention the instruction tuning and RLHF stages), which are extremely compute-expensive and out of scope of this paper.
>
> **Word count and word index are not exchangeable in some contexts. It is better to make this more concise. For example, in the abstract "However, current PE methods use token counts to derive position," I suppose this should be "token index" instead of "token count"**
>
> Index usually refers to the absolute position of things. However, the current PE methods like RoPE measure word count between the source and target words, this word count is more suited terminology.
>
> **Visualizations should be consistent. The style of Figure 5 is different from others.**
>
> Sure, we can fix that.
>
> **Have you tried your pretrained model for language understanding task?**
>
> Yes, we applied our pretrained model to language understanding tasks and presented the result in Table 12.

---

> ### Comment · Reviewer_XPkW · 2024-11-25
> **Thanks for clarification**
>
> W1: The major concern is the novelty. The learnable vector or contextual position is based on the idea of position interpolation and fractional positional encoding explored in Insert Transformer, particularly in [1].
>
> Let me give you an example. Suppose we need to add PE to a sequence: a b c d e f g.  I will do this step-by-step based on my understanding.
>
> CoPE:
>
> Step 1 (1st layer):
> - seq:   a       b       c        d       e        f       g
> - PE  :         1    1/2      2         3       4        5      6    (sigmoid-> integers , interpolation> fractional values)
>
> Step 2 (e.g., 2nd layer):
> - seq:   a       b       c        d       e        f       g
> - PE:    1    1/5      2/5      3/5   4/5      2     3    (refresh)
>
> What insertion Transformer do:
>
> Step 1 (1st decoding):
> - seq:   a     d
> - PE:       1    2     (relative or absolute PE)
>
> Step 2 (2nd decoding):
> - seq:   a       b       d     g
> - PE:    1    2       3     4    (refresh integers based on context)
>
> insertion Transformer + [1]:
>
> Step 1 (1st decoding):
> - seq:   a     d
> - PE:       1    2
>
> Step 2 (2nd decoding):
> - seq:        a     b    c       d     g
> - PE      :    1    1/3 2/3     2     3    (do interpolation if the neighboring positions exist.  Note that you can also "refresh all integers based on context", similar to insertion Transformer .)
>
>
> As presented in this toy example, the method significantly overlaps with the insertion Transformer family. The sigmoid value idea is also shared with relative PE. Please point out if I have substantial misunderstandings.

---

> > ### Author Response · Authors · 2024-11-26
> > **Thank you for your time**
> >
> > **As presented in this toy example, the method significantly overlaps with the insertion Transformer family.**
> >
> > Thank you for providing a detailed example! There seems to be a major misunderstanding of our method. While both CoPE and Insert transfomer use fractional positions, that is not the main objective of CoPE. The goal of CoPE is to make positions change depending on context, i.e. actually words themselves. The above example does not capture this property at all. For example, CoPE can increment positions only on token “a”. So given two sequences
> >
> > “a b c a d a a e f” and “b c a a d e a a f”,
> >
> > their positions (when queried from the last token) will be different
> >
> > “4 3 3 3 2 2 1 0 0” and “4 4 4 3 2 2 2 1 0”
> >
> > Insert transformer cannot perform such context dependent position encoding. Both sequences will have the same positions, assuming tokens are inserted in the same order. Let us explain each approach in more detail.
> >
> > Insert Transfomer relies on a different generation strategy, where tokens can be inserted before previously generated tokens, and authors propose a fix in PE when the absolute position of a token may change. They simply interpolate PE for the new token, citing from the paper: “Whenever a new token wnew is inserted between two existing tokens $w_{left}$ and $w_{right}$, its positional representations will be calculated with a function $f$ applying to its current left and right neighbors: $p_{new} = f(p_{left}, p_{right})$. … The function $f$ is modeled by a linear layer1 which takes the concatenation of the two neighbors’ positional embeddings and outputs a new vector of the model size.” This approach is specific for insertion-style transformers, and is not suitable for regular left-to-right transformers (because there is nothing to interpolate between).
> >
> > Our approach on the contrary is developed for regular, left-to-right transformers. Unlike Insert Transformer, our position encodings depend on *context*, i.e. on attention scores (eq-ns 3 and 4), where fractions are not enforced as interpolation between neighboring positions as in Insert Transformer, but computed based on attention scores (eq-ns 3 and 4). What example you have provided is missing, is the attention-based (or context-dependent) encodings computation, which is a key idea of the paper.
> >
> > **The sigmoid value idea is also shared with relative PE**
> >
> > Can you please clarify this more because this doesn’t sound correct? Sigmoid operation is not used anywhere in the relative PE baseline. Relative PE simply counts tokens and assigns embedding to each (whole) number. Our method of using contextual sigmoid gating applied to query and key dot products is novel and to the best of our knowledge never used before in position encoding methods. The use of context in position encoding is the heart of our approach.

---

> ### Comment · Reviewer_XPkW · 2024-12-02
> **Sorry for late reply**
>
> 1. To be clear, I want to use my toy example to demonstrate that recomputing PE based on context is not a new angle. As you can see, the context is changed at each step in Insertion Transformers because of newly inserted words.
> 2. Thanks for clarifying. Now, the method is clearer to me, and I can raise my score.
>
> ----------------------------New Review----------------------------
>
> Now, I have some new questions.
>
> Based on my understanding. the p_max is an important component. However, from Table 9, we can see even a small p_max can work well. This is unconvincing or counterintuitive because a small p_max will produce the same PE for many positions. In this case, CoPE can not distinguish positions. Do you have any other experiments or insights for this? (Note that, since the method uses contextualized representations, the anisotropic problem will make the attention scores even in deep layers, meaning gate values are very similar across all positions.)

---

> > ### Author Response · Authors · 2024-12-03
> >
> > We would like to thank you for your re-consideration of our work, please find the answers to your follow-up questions below:
> >
> > **the context is changed at each step in Insertion Transformers because of newly inserted words.**
> >
> > We would like to highlight that even though the context is changing, the position encodings in the insertion transformer are NOT context-dependent – they are still position-dependent (and insertion ordering dependent). For instance, in your example it does not matter if the newly inserted token is “cat” or “dog”, and the only thing that matters is where that token is inserted. On the contrary, CoPE relies on attention scores, which depend both on key and query values, thus can count “cat” and “dog” differently.
> >
> > **Based on my understanding. the p_max is an important component. However, from Table 9, we can see even a small p_max can work well**
> >
> > The parameter p_max defines the maximum possible position count we use. This is useful for efficiency, but the main conceptual contribution of the paper (the main idea of contextual position encodings) is not particularly dependent on it – one can always use the full length, i.e. p_max = T, which tends to work well.
> >
> > For a particular task, one may be able to choose small p_max and still obtain similar performance results – but more efficiently. For example, if the gates are sparsely activated (e.g. counting sentences), one can cover the whole context T with only a few positions, in contrast to counting tokens (as required in something like RoPE).
> >
> > Yes, a small p_max value worked well for the wikitext experiment as shown in Table 9. However, our larger scale experiment showed that larger p_max is necessary, probably because larger models use longer context with more relevant fine-grained positioning information being necessary. We did  perform an additional small ablation study on a 1.4B model, comparing CoPE and CoPE+RoPE with different p_max values, but didn’t include it in the paper, as these experiments were run on a small percent of total training steps. Below we provide the results of these ablations. In summary, in our experiment increasing p_max leads to improvements in performance, but also increases memory consumption.
> >
> > CoPE + RoPE, p_max = 128* corresponds to the model reported in the paper, and * means that CoPE was added to every 6th layer only. We report validation perplexity.
> >
> > _________________________________| RoPE | CoPE + RoPE, p_max = 128* | CoPE + RoPE, p_max = 64* | CoPE + RoPE, p_max = 128 | CoPE + RoPE, p_max = 64 | CoPE + RoPE, p_max = 128* + sep key
> >
> > Perplexity, Wikipedia, 2% of train tokens:  | **7.374** | 7.416 | 7.427 | 7.694 | 7.662 | 7.492
> >
> > Perplexity, Wikipedia, 11% of train tokens:| 6.595 | 6.399 |  **6.373** | 6.658 | 6.629 | 6.419
> >
> > Perplexity, Wikipedia, 53% of train tokens:| 6.006 | **5.787** |  5.832 | - | - | -
> >
> > Perplexity, C4, 2% of train tokens:             | **11.473** | 11.667 | 11.655 | 11.993 | 11.906 | 11.659
> >
> > Perplexity, C4, 11% of train tokens:           | 10.397 | 10.161 | **10.146** | 10.565 | 10.444 | 10.182
> >
> > Perplexity, C4, 53% of train tokens:           | 9.56 | **9.307** | 9.316 | - | - | -
> >
> >
> > **since the method uses contextualized representations, the anisotropic problem will make the attention scores even in deep layers, meaning gate values are very similar across all positions.**
> >
> > We are not sure if “attention scores even in deep layers” actually occurs in our trained models. If it is occurring, then this means attention maps will be flatter (not sharp) at higher layers, which is likely to negatively affect the model’s performance regardless of the position encoding. In any case, it doesn’t seem to hinder CoPE’s capability, as our experiments demonstrated that CoPE works well in practice.

---

### Comment · Area_Chair_rGvP · 2024-11-25
**Respond  to the author's rebuttals**

Dear Reviewers,

Thank you for your efforts in reviewing this paper. We strongly encourage you to review and respond to the author's comments to promote a more dynamic exchange of ideas.

Thank you for your collaboration.


Best regards,
ICLR 2025 Area Chair

---

### Meta-Review · Area_Chair_rGvP · 2024-12-20

**Metareview:**

The paper proposes a novel position encoding method, Contextual Position Encoding (CoPE),  to enhance transformer-based language models by integrating context into position addressing. This approach aims to overcome the limitations of existing context-independent PE methods, particularly in tasks involving word and sentence counting. Experimental results on the Flip-Flop language modeling task, the selective copy task, the symbolic counting task, the WordCount task, the language modeling task, and the code modeling task demonstrate the effectiveness of the proposed CoPE.

However, there are noteworthy concerns. 1) the novelty of the proposed CoPE; 2) the details regarding the model design; and 3) the computational cost associated with CoPE.  Addressing the aforementioned questions will render the paper more comprehensive and impactful.

**Additional Comments On Reviewer Discussion:**

1) the novelty of the proposed CoPE.
2) the details regarding the model design.
3) the computational cost associated with CoPE.

The responses to questions 2 and 3 did not alleviate the reviewers' doubts.

---

### Decision · Program_Chairs · 2025-01-22

Reject